

# Eatify
## System zarządzania restauracją

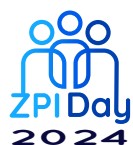

**Autorzy**: Michał Klatkowski ⦿ · Aleksandra Pluta ⦿ · Mikita Sakirko ⦿ · Kacper Wieczorek ⦿

**Opiekun:** Adrianna Kozierkiewicz

### Streszczenie

Celem projektu Eatify było stworzenie kompleksowej platformy do zarządzania restauracją. Aplikacja obejmuje szeroki wycinek rzeczywistości, na który składają się perpektywy użytkowników takich jak: klient, menadżer, kucharze oraz kelnerzy. Wynikiem pracy zespołu jest aplikacja webowa, która pozwala na składanie zamówienia oraz rezerwacji, a także zarządzanie funkcjonowaniem restauracji przez pracowników w zależności od ich roli. Projekt niesie ze sobą gotowe do wdrożenia rozwiązanie, które w łatwy sposób może zostać przystosowane do potrzeb konkretnej restauracji, pozwala na dostowanie komponentów takich jak menu, układ sali, czy zarządzanie personelem. Aplikacja jest odpowiednim rozwiązaniem dla restauracji, które pragną zmodernizować swój system na taki, który w intuicyjny sposób pozwoli zarządzać znaczną częścią funkcji lokalu gastronomicznego.

## 1 ROZWÓJ

### 1.1 Wstęp

W obecnym świecie często można napotkać nowe lokale, które zaczynają swoją działalność w branży gastronomicznej. Aby te mogły być konkurencyjne na rynku, powinny one posiadać zmodernizowany systemu zarządzania restauracją. który zaoferuje funkcjonalności takie jak obsługa klientów, zarządzanie pracownikami, stanem posiadania składników i stanem procesów zachodzących w trakcie realizacji zamówień. Posiadanie na własność przez restaurację systemu, który pozwoliłby rozwiązać problem zamawiania jedzenia, i dodatkowo wspierałby także restaurację w zakresie kadry pracowniczej, pozwoliłoby w pełnym wymiarze odciążyć właściciela z konieczności fizycznego monitorowania pracy personelu. Ułatwiłoby to także w znaczny sposób kontrolę menu, z którego klienci mogliby zamawiać posiłki, oraz pozwoliłoby na uproszczenie zarządzania rezerwacjami, które w restauracjach bez dedykowanego systemu często są przeprowadzane telefonicznie.

Wyżej wspomniane zintegrowane środowisko dla restauracji jest oferowane przez aplikację Eatify. System ten łączy w sobie cechy obecnie znanych firm dowożących jedzenie, w zakresie składania zamówienia (w tym także płatności online), a także dokłada funkcjonalności związane z działalnością fizycznej jednostki restauracji. Podstawową funkcjonalnością związaną z tym jest składanie rezerwacji, które pozwoli m.in. na dobrór odpowiadającego stolika ze schematu sali. Personalizacja konta pod kątem diet oraz nietolerancji, pozwoli w łatwiejszy dla klienta sposób wyszukiwać posiłki, które są do niego przystosowane. Informowanie kucharzy o posiłkach, które są do przygotowania, oraz informacja dla kelnerów o posiłkach do dostarczenia pozwala usprawnić przepływ wiedzy związanej ze stanem zamówienia. Właściciel, poza manipulacją menu, dietami oraz pomieszczeniami i rozłożeniem stolików, posiada opcję generowania raportów, pozwalającą na szybkie uzykanie informacji np. o przychodach, zamawianych posiłkach albo składanych rezerwacjach.

### 1.2 Prace związane z tematem

Na rynku gastronomicznym istnieje wiele rozwiązań wspierających zarządzanie restauracjami, jednak znaczna większość z nich skupia się na wybranej części funkcjonalności związanej z kierownictwem, bądź oferuje rozwiązania dedykowane dla większych sieci. W tej sekcji zostanie przeanalizowana część rynku, która odpowiada za realizację wcześniej wspomnianych funkcjonalności. Sekcję tę można rozdzielić na dwa segmenty na temat analizy rozwiązań:

· dostawców jedzenia

· dużych sieci gastronomicznych.

Najczęściej wykorzystywaną ofertą na rynku lokali gastronomicznych są usługi dostawców jedzenia. Aby zaspokoić część potrzeb, właściciele lokali gastronomicznych często decydują się na współpracę dostawcami jedzenia, jednak mimo tego, że rozwiązanie to zyskuje popularność w ostatnich latach, nie rozwiązuje ono w pełni problemów, z którymi borykają się właściciele. Odciążenie restauracji w zakresie zamawiania jedzenia przekłada się na duże koszty związane z prowizją pobieraną przez firmy dowozowe. Najpopularniejszymi z nich są: **Glovo, Pyszne.pl, Uber Eats** oraz **Wolt**. Według najnowszego raportu firmy Stava, zajmującej się m.in. analizą rynkową branży "delivery", tylko 25% zamówień online odbywało się poprzez własne kanały zamówień, przy 75% zamówień przeprowadzanych przez zewnętrznych dostawców [1]. Niestety, nie jest to kompleksowa odpowiedź na potrzeby właścicieli restauracji w zakresie zarządzania. Przerzucenie funkcjonalności składania zamówień na zewnętrzne firmy wiąże się ze sporymi kosztami, które są dużym obciążeniem dla właścicieli małych lokali, w szczególności tych, które dopiero zaczynają działania w branży, i chcą rozwinąć swoją działalność. Na przykładzie firmy Glovo, koszt usługi dostawy zamówienia to 30% wartości zamówienia, a sama funkcjonalność składania zamówienia z odbiorem na miejscu wiąże się z kosztem 15% wartości zamówienia. [2]. Te koszta, w powiązaniu z innymi kosztami działalności lokalu, mogą w znaczny sposób utrudniać rozwój restauracji.

Na rynku w codziennym życiu często można napotkać konkurencyjne rozwiązania, które są wykorzystywane przez duże korporacje. Każda większa sieć lokali gastronomicznych posiada swoją aplikację, bądź stronę internetową, która pozwala sprawnie zarządzać pracą danego lokalu. Gigantyczne firmy, takie jak **McDonald's, KFC, Pizza Hut** czy **Burger King** posiadają powszechnie znane systemy, które od strony klienta pozwalają na złożenie zamówienia, oraz dostarczają ułatwiają zarządzanie pracą personelu. Należy jednak pamiętać, że wspomniane systemy są dedykowane dla konkretnych firm, i nie są oferowane komercyjnie dla innych restauracji.

Aplikacja Eatify jest odpowiedzią na potrzeby restauracji, które nie mogły być zaspokojone przez wcześniej wspomniane rozwiązania. W porównaniu do wspomnianych wcześniej usług, jest to aplikacja która w kompleksowy sposób - a nie częściowo - udostępnia wiele funkcjonalności, takie jak np. rezerwacja lub możliwości personalizacji ustawienia stolików, które nie występują często w konkurencyjnych systemach.

Efektem ośmiu tygodni pracy 4-osobowego zespołu jest wcześniej wspomniana aplikacja Eatify. Została ona wytworzona przy wsparciu dobrze znanych technologii - Spring Boot, React oraz AWS, które oferują długoterminowe wsparcie. Był to ważny czynnik doboru technologii, ponieważ głównym założeniem projektowym była możliwość utrzymania aplikacji na wiele miesięcy. Planowane było także poświęcenie znacznej ilości czasu na testowanie oraz udokumentowanie aplikacji, aby ta była wysoko jakościowym produktem. Założeniem finansowym projektu było maksymalnie ograniczenie kosztów wytwarzania oprogramowania, poprzez wykorzystanie zasobów takich jak dostęp do platformy AWS w ramach planu studenckiego.

## 1.3  Wyniki

Wynikiem prac zespołu jest aplikacja, która odpowiada na realne potrzeby właścicieli lokali gastronomicznych. Łączy ona ze sobą funkcjonalności zamawiania jedzenia z zarządzaniem restauracją na wielu poziomach. Obsługuje ona zarówno zamówienia na wynos, jak i na miejscu - z opcją płatności on-line bądź gotówkowej.

Najważniejszym użytkownikiem systemu jest klient. Może on - tak samo jak w konkurencyjnych rozwiązaniach - złożyć zamówienie poprzedzone płatnością on-line. Jest on w stanie również złożyć rezerwację, korzystając przy tym z wyboru stolika ze schematu, który został odpowiednio wcześniej skonfigurowany przez menadżera. System uwzględnia również potrzeby klientów, którzy posiadają nietolerancje - personalizacja konta w kontekście pozwala w łatwy sposób odfiltrować propozycje posiłków.

Ze strony menadżera, system oferuje możliwość personalizacji konfiguracji restauracji. W swoim panelu, właściciel jest w stanie dodawać posiłki, diety, składniki i ich widocznosc w menu. Może określać, czy dany składnik może zostać zmodyfikowany przez użytkownika, tj. dodany lub usunięty z zamawianego posiłku. W odniesieniu do wcześniej wspomnianej rezerwacji, menadżer posiada dedykowany panel do konfigurowania rozłożenia stolików na schemacie sali. Do analizy wyników sprzedaży, rezerwacji oraz innych danych, właściciel może wygenerować raporty w formacie PDF. W kontekście zarządzania personelem, może on tworzyć konta pracownikom.

W skład wcześniej wspomnianego personelu składają się kelnerzy oraz kucharze. Kelnerzy są w stanie monitorować stan zamówień na każdym etapie jego realizacji. Mają możliwość zarówno złożenia zamówienia, gdy to jest realizowane na miejscu, jak i śledzenia stanu zamówienia. Mogą oni przypisać się do danego zamówienia, które jest do dostarczenia, lub obsłużenia na miejscu. Z kolei kucharze w ramach systemu mają dostęp do przeglądania zamówień które zostały przekierowane do realizacji, oraz do przekazania zamówienia jako przygotowane do kelnerów. Mogą oni także zmieniać na bieżąco dostępność danych składników, która implikuje dostępność posiłków w menu.

Cała aplikacja jest skalowalna oraz łatwa do przystosowania do potrzeb poszczególnej restauracji, przez co może być reużywana w wielu firmach świadczących usługi gastronomiczne. Aplikacja spełnia założone na początku wymagania funkcjonalne oraz biznesowe, a także jest zabezpieczona w odpowiedni sposób, który bazuje na systemie roli, jaką w systemie posiada użytkownik. Zastosowanie silnych metod szyfrowania haseł i wrażliwych danych użytkowników takie jak enkrypcja algorytmem BCrypt, zasalanie hasła oraz skorzystanie z usług zewnętrznych dostawców tożsamości takich jak Google czy Facebook powoduje, że nawet w przypadku nieautoryzowanego dostępu do bazy danych, wrażliwe dane nie zostaną wykradzione.

Aplikacja wyróżnia się zastosowaniem algorytmów rekomendacji. Te zostały zaimplementowane zarówno do propozycji posiłków przy zamówieniach, oraz do propozycji wyboru miejsc przy rezerwacjach. Pozwalają one na lepsze dostosowanie wspomnianych elementów do preferencji klienta. Na szczególną uwagę zasługuje zastosowanie **algorytmu apriori** do rekomendacji posiłków, bazując na globalnej historii zamówień w restauracji. Pozwala on wygenerować reguły asocjacyjne, które ze wskazaną pewnością określają jak często występują określone relacje między danymi zamawianymi posiłkami. Dzięki temu rozwiązaniu aplikacja jest w stanie dawać rekomendacje, które są oparte na faktycznych danych zamówień.

Rozwiązaniem ukierunkowanym na klienta jest także zastosowanie kodów QR przypisanych do poszczególnych stolików. Pozwoli to na jeszcze szybsze złożenie zamówienia do stolika w przypadku zamówienia przez urządzenie elektroniczne. Klient w takim wypadku nie będzie musiał ręcznie wybierać ze schematu stolika przy którym usiadł, lub wpisywać jego numeru,

Tak jak zostało to wspomniane wcześniej, Eatify pozwala obsłużyć płatności on-line poprzez usługi zewnętrznego dostawcy płatności jakim jest **PayU**. Głównym powodem, z jakiego został wybrany było silne bezpieczeństwo transakcji jakie oferuje. Dodatkowo, jest to polski znany dostawca, który zyskał popularność w wielu rozwiązaniach, aplikacjach i platformach. Według raportu o e-handlu w Polsce, aż 46% systemów posiadających płatność on-line, korzysta z usług tego dostawcy [3].

Cały system posiada także skonfigurowany system mailowy, który powiadamia klientów o rejestracji, złożonej rezerwacji czy zmianie hasła. Jest to podstawowy element komunikacji z klientem, dlatego został on uwzględniony i zaprojektowany już na samym początku projektu.

Projekt posiada dedykowaną infrastrukturę chmurą. To rozwiązanie powoduje, że aplikacja staje się bardzo łatwa do wdrożenia i utrzymania. Zautomatyzowane skrypty pozwalają w ciągu krótkiego czasu przygotować aplikację do pełnego publicznego udostępnienia. Przerzucenie odpowiedzialności działania aplikacji na zewnętrzne serwisy chmurowe spowodowało, że w bezproblemowy sposób można zarządzać różnymi elementami aplikacji, pozwalając na skalowanie poszczególnych serwisów, dodanie nowych w miarę potrzeby i zarządzanie nimi w zorganizowany sposób. Schemat architektury został przedstawiony na diagramie poniżej 1

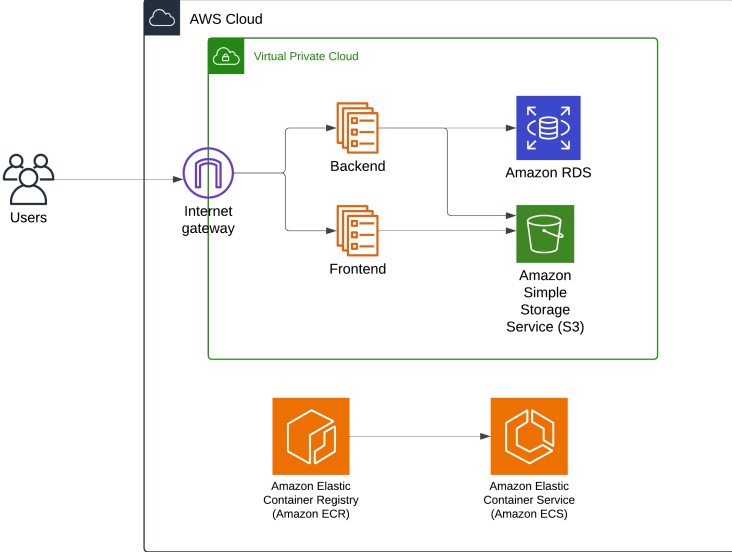

Rysunek 1: Diagram architektury chmurowej platformy Eatify

## 2   WNIOSKI

### 2.1   Wnioski

Platforma Eatify w znacznym stopniu realizuje założenia przedstawione przed rozpoczęciem prac, i zawiera większość funkcjonalności, które zostały zaplanowane. Aplikacja jest gotowa do wdrożenia w celach komercyjnych tak, aby w łatwy i szybki sposób dostosować konfigurację restauracji do stanu faktycznego w jakim się znajduje. Jest ona gotowa obsłużyć wszystkich użytkowników systemu tak, aby wspomóc ich pracę bądź korzystanie z usług restaurauracji.

Platforma jest konkurencyjnym produktem, w porównaniu do innych rozwiązań obecnych wszechobecnie na rynku branży gastronomicznej. Jej kompleksowość, łatwość obsługi i wdrożenia powoduje, że może być atrakcyjnym produktem dla potencjalnych nabywców, który odciąży właścicieli w kontekście kosztów i nakładu pracy w ramach zarządzania restauracją.

Skalowalność produktu która jest oferowana przez infrastrukturę chmurową powoduje, że produkt jest w stanie obsłużyć zarówno małe lokale gastronomiczne, jak i duże restauracje, co powoduje, że rozwiązanie Eatify jest odpowiednim produktem dla każdego odbiorcy, niezależnie od wielkości i skali prowadzonej działalności.

### 2.2   Kierunki rozwoju

Aplikacja mimo tego, że jest zrealizowana pod kątem założonych funkcjonalności i jest zdatna do użytku, posiada pewne płaszczyzny, w których można by było ją rozwijać.

Pierwszym pomysłem, który zrodził się już w czasie planowania funkcjonalności i zakresu projektu, była implementacja chatbota opartego o mechanizmy sztucznej inteligencji. Z założenia, ten miałby przetwarzać wiadomość od klienta. W taki oto sposób, klient mógłby się dowiedzieć rzeczy na temat menu, stanu sali na dany dzień, najczęściej zamawianych daniach, i innych zagadnieniach, które mogłyby interesować użytkownika, a nie byłyby predefiniowanymi funkcjami platformy. Realizacja chatbota nie była możliwa w czasie trwania projektu, z racji na czasochłonność takiego rozwiązania.

Kolejnym pomysłem, który mógłby zostać zrealizowany w ramach tego projektu, byłby panel personalizacji witryny przez managera. Aplikacja mogłaby udostępniać dedykowane wbudowane narzędzie do: personalizacji poszczególnych stron, komponentów, kolorystyki bądź rozmieszczenia kafelków i sekcji na stronie. Jest to ważny element platformy która z założenia ma być uniwersalnym szkieletem aplikacji dla restauracji. Często restauracja bądź sieć lokali gastronomicznych bazuje swój marketing na design'ie marki, dlatego ważnym elementem powinno być dostosowanie strony do szaty graficznej firmy.

W odniesieniu do obecnie udostępnianych na rynku usług, planowane jest także dostosowanie się do obecnych realiów i obsłużenie możliwości dostawy posiłków, w ramach dostawy na wybrany adres. Udostępnienie takiej funkcjonalności pozwoliłoby w pełni uniezależnić właścicieli lokali gastronomicznych od firm zajmujących się dostarczaniem jedzenia, i tym samym zaoszczędzić znaczną ilość środków, które zostałyby zużytkowane na usługi zewnętrznętrznego dostawcy.

Ważną funkcjonalnością, która nie znalazła się w pierwszej wersji aplikacji jest generowanie faktur za zamówienia posiłków. Jest to ważny aspekt płatności, przykładowo przy rozliczeniach zamówień na koszt firmy. Z racji na niski priorytet tej funkcjonalności w kontekście innych funkcji jakie oferuje platforma, fakturowanie nie znalazło się na liście zadań do zrealizowania, jednak jest to ważny czynnik w zakresie obsługi płatności.

Ważnym elementem projektowania i wdrażania aplikacji webowych, jest ich zabezpieczenie. Zgodnie ze wcześniej wspomnianymi wynikami, aplikacja jest zabezpieczona wewnątrz na wiele sposobów, które powodują, że dane znajdujące się na niej są bezpiecznie użytkowane. Jednakże, z racji na wspomniane ograniczenia studenckiej subskrybcji AWS, nie było możliwości uzyskać stałej domeny dla naszej aplikacji. To skutkowało dużymi problemami z uzyskaniem certyfikatu SSL, a tym samym zmianą protokołu z HTTP na HTTPS. W przypadku wdrożenia aplikacji w celu komercyjnym, na subskrypcji innej niż studencka, jest to priorytet do uwzględnienia.

### 2.3   Podziękowania

Jako zespół, pragniemy podziękować naszej Pani Promotor **dr hab. inż. Adriannie Kozierkiewicz** za nieocenione wsparcie w trakcie realizacji projektu - zarówno przed jego samym rozpoczęciem, na etapie projektowania, jak i w jego trakcie. Pragniemy także podziękować Pani **dr inż. Bogumile Hnatkowskiej**, za cenne wsparcie przy procesie wytwarzania oprogramowania, które pozwoliło nam w znaczny sposób poprawić jakość naszego rozwiązania. Podziękowania kierujemy także w stronę **dr inż. Marcina Pietranika**, który udzielił nam wsparcia w zakresie architektury aplikacji i zaproponował rozwiązania, które pojawiły się w ostatecznej wersji aplikacji.

## LITERATURA

[1] Stava. Raport o rynku dowozów jedzenia w Polsce. `https://stava-reports.s3.eu-central-1.amazonaws.com/Raport+Stava+o+rynku+dowozow+jedzenia+2024.pdf`, 2024.

[2] Glovo. Koszty usług firm dowozowych na przykładzie Glovo. `https://sell.glovoapp.com/pl/pl/services/pricing/`.

[3] PayU. Raport o e-handlu w Polsce "E-commerce w Polsce 2021". `https://poland.payu.com/blog/najnowsze-dane-o-e-handlu-w-polsce-przeczytaj-raport/`, 2021.
