# OpenReview forum: "Eatify - system zarządzania restauracją"
_pwr.edu.pl/Wrocław_University_of_Science_and_Technology/2024/ZPI_Day — Wrocław University of Science and Technology 2024 ZPI Day Submission_

### Official Review · Reviewer_n8cH · 2024-12-04
**Eatify - system zarządzania restauracją - ocena ogólna 5**

**Confidence:** 5
**Significance Of Results:** 5
**Overall Quality:** 5

**Compliance With Template:**

5: Very High Quality – The article contains all the required sections, which are written in a very detailed, clear, and error-free manner. The structure is professional and meets expectations, and the content adheres to the highest substantive and formal standards.

**Description Of Results:**

5: Very High Quality – The results are described in detail, clearly and comprehensively, supported by thorough evaluation, analysis, and convincing usage examples. The description meets the highest substantive standards.

**Feedback On Consistency:**

Praca ogólnie poprawna. Zauważone błędy:
- żargon (np. stworzenie)
- niewłaściwe stosowanie pojęcia funkcjonalność
- niewłaściwe sformułowanie: "Najważniejszym użytkownikiem systemu jest klient"
- błędy stylistyczne, gramatyczne, interpunkcyjne - niewłaściwe umieszczanie przecinków (ich nadmiar lub brak), np. "z kolei kucharze w ramach systemu mają dostęp do przeglądania zamówień które zostały przekierowane do realizacji, oraz do przekazania zamówienia jako przygotowane do kelnerów.", "Platforma jest konkurencyjnym produktem, w porównaniu do innych rozwiązań obecnych wszechobecnie
na rynku branży gastronomicznej.", "Często restauracja bądź sieć lokali gastronomicznych bazuje swój marketing na design’ie
marki, dlatego ważnym elementem powinno być dostosowanie strony do szaty graficznej firmy"
- niewłaściwe odwołanie do rysunku 1
- pozostawianie "wdów" czyli pojedynczych słów w ostatnim wierszu akapitu
- pozostawianie "sierot" czyli pojedynczej litery na końcu wiersza
- przypisywanie rzeczom cech ludzkich, np. "aplikacja jest w stanie dawać rekomendacje"
- w spisie literatury przy adresach stron internetowych warto podać datę zamieszczenia artykułu, datę dostępu do strony.

**Potential For Development:**

Opisano kierunki rozwoju

**Project Nature Evaluation:**

Zespołowy projekt inżynierski

**Technical Language Precision:**

4: High Quality – The language is appropriate for a technical report. Terminology is used correctly, and statements are precise, with only minor shortcomings that do not affect the overall clarity.

---

### Official Review · Reviewer_FNLq · 2024-12-05
**Eatify - system zarządzania restauracją**

**Confidence:** 5
**Significance Of Results:** 4
**Overall Quality:** 4

**Compliance With Template:**

5: Very High Quality – The article contains all the required sections, which are written in a very detailed, clear, and error-free manner. The structure is professional and meets expectations, and the content adheres to the highest substantive and formal standards.

**Description Of Results:**

4: High Quality – The results are described in detail and supported by usage examples or evaluations. The description is reliable but may lack full depth of analysis.

**Feedback On Consistency:**

EN:
The project description for Eatify is generally consistent and logically structured. It begins with a clear problem statement, identifying the challenges faced by small and independent restaurants in managing their operations without relying on costly third-party services or proprietary systems designed for large chains (KFC, etc.). The presentation of results aligns well with the initial objectives, detailing how Eatify addresses these challenges through features like online ordering, reservations, staff management, and personalized dietary options. The conclusions effectively summarize the project’s achievements and its potential impact on modernizing restaurant operations.

However, the description lacks a thorough market analysis, such as a SWOT analysis, which would provide a deeper understanding of the competitive landscape and the specific needs of potential users. Including this information would strengthen the logical flow of the project by justifying the need for Eatify and highlighting its advantages over existing solutions. Additionally, while the recommendation system is mentioned, there’s limited discussion on how it considers both customer preferences and the restaurant’s current capabilities, which is crucial for practical applicability.

PL:
Opis projektu Eatify jest ogólnie spójny i logicznie skonstruowany. Rozpoczyna się od jasnego określenia problemu, identyfikując wyzwania stojące przed małymi i niezależnymi restauracjami w zarządzaniu ich działalnością bez polegania na kosztownych usługach zewnętrznych lub zastrzeżonych systemach zaprojektowanych dla dużych sieci (np. KFC). Prezentacja wyników jest zgodna z początkowymi celami, szczegółowo opisując, w jaki sposób Eatify radzi sobie z tymi wyzwaniami dzięki funkcjom takim jak zamówienia online, rezerwacje, zarządzanie personelem i spersonalizowane opcje dietetyczne. Wnioski skutecznie podsumowują osiągnięcia projektu i jego potencjalny wpływ na modernizację operacji restauracyjnych.

W opisie brakuje jednak dokładnej analizy rynku, takiej jak analiza SWOT lub podobna, która zapewniłaby głębsze zrozumienie konkurencyjnego krajobrazu i konkretnych potrzeb potencjalnych użytkowników. Uwzględnienie tych informacji wzmocniłoby logiczny przepływ projektu poprzez uzasadnienie potrzeby Eatify i podkreślenie jego przewagi nad istniejącymi rozwiązaniami. Ponadto, chociaż wspomina się o systemie rekomendacji, dyskusja na temat tego, w jaki sposób uwzględnia on zarówno preferencje klientów, jak i obecne możliwości restauracji, jest ograniczona, co ma kluczowe znaczenie dla praktycznego zastosowania.

**Potential For Development:**

EN:
The article indicates significant possibilities for further work and practical applications of Eatify’s results. Future developments could include:
- Native Delivery Service: Implementing an integrated delivery system to reduce reliance on third-party services, potentially lowering costs and improving customer service.
- Invoice Generation and Financial Tools: Adding features for automatic invoice generation and enhanced financial reporting would benefit both the restaurant and its customers, particularly for corporate clients.
- AI Chatbot Integration: Developing an AI-powered chatbot to handle customer inquiries about menu items, availability, and reservations can improve customer engagement and reduce the workload on staff.
- Security Enhancements: Addressing the current limitation regarding SSL certification to support HTTPS protocols will enhance data security and customer trust.
- Customization and Branding: Allowing restaurants to customize the platform’s UI and branding can make the system more appealing to a wider range of establishments, enhancing marketability.

Focusing on these areas can expand Eatify's functionality and adaptability, making it a more robust and appealing solution for the restaurant industry. These developments not only address current limitations but also open avenues for innovation, potentially setting Eatify apart from competitors and meeting evolving market demands.

PL:
Artykuł wskazuje na znaczące możliwości dalszej pracy i praktycznych zastosowań wyników Eatify. Przyszły rozwój może obejmować:
- Natywna usługa dostawy: Wdrożenie zintegrowanego systemu dostaw w celu zmniejszenia zależności od usług zewnętrznych, potencjalnie obniżając koszty i poprawiając obsługę klienta.
- Generowanie faktur i narzędzia finansowe: Dodanie funkcji automatycznego generowania faktur i ulepszonego raportowania finansowego przyniosłoby korzyści zarówno restauracji, jak i jej klientom, szczególnie klientom korporacyjnym.
- Integracja z chatbotem AI: Opracowanie chatbota opartego na sztucznej inteligencji do obsługi zapytań klientów dotyczących pozycji menu, dostępności i rezerwacji może poprawić zaangażowanie klientów i zmniejszyć obciążenie personelu.
- Ulepszenia bezpieczeństwa: Rozwiązanie obecnego ograniczenia dotyczącego certyfikacji SSL w celu obsługi protokołów HTTPS zwiększy bezpieczeństwo danych i zaufanie klientów.
- Personalizacja i branding: Umożliwienie restauracjom dostosowania interfejsu użytkownika i marki platformy może sprawić, że system będzie bardziej atrakcyjny dla szerszej gamy lokali, zwiększając atrakcyjność rynkową.

Koncentrując się na tych obszarach, Eatify może rozszerzyć swoją funkcjonalność i zdolność adaptacji, czyniąc ją bardziej solidnym i atrakcyjnym rozwiązaniem dla branży restauracyjnej. Zmiany te nie tylko rozwiązują obecne ograniczenia, ale także otwierają drogę do innowacji, potencjalnie odróżniając Eatify od konkurencji i spełniając zmieniające się wymagania rynku.

**Project Nature Evaluation:**

EN:
Eatify exhibits strong characteristics of engineering work. It employs advanced technical methods and modern technological solutions to address real-world problems in restaurant management. The use of React for the frontend ensures a responsive and user-friendly interface, while Spring Boot on the backend provides reliability and scalability. The integration with AWS for cloud storage and hosting demonstrates an understanding of building scalable, cloud-based applications.

The application of the "Apriori" algorithm for personalized recommendations showcases the use of artificial intelligence to enhance customer experience. However, to fully leverage this technology, the recommendation system should consider customer preferences and the restaurant’s current inventory and resource availability. This would ensure that recommendations are feasible and optimize operational efficiency. The project effectively combines various technical components to create a comprehensive management system, reflecting the utility and innovation expected in engineering solutions.

PL:
Eatify wykazuje silne cechy pracy inżynierskiej. Wykorzystuje zaawansowane metody techniczne i nowoczesne rozwiązania technologiczne do rozwiązywania rzeczywistych problemów w zarządzaniu restauracjami. Wykorzystanie React dla frontendu zapewnia responsywny i przyjazny dla użytkownika interfejs, podczas gdy Spring Boot na zapleczu zapewnia niezawodność i skalowalność. Integracja z AWS dla przechowywania danych w chmurze i hostingu pokazuje zrozumienie budowania skalowalnych aplikacji opartych na chmurze.

Zastosowanie algorytmu "Apriori" do spersonalizowanych rekomendacji pokazuje wykorzystanie sztucznej inteligencji w celu poprawy jakości obsługi klienta. Aby jednak w pełni wykorzystać tę technologię, system rekomendacji powinien uwzględniać preferencje klientów oraz bieżące zapasy i dostępność zasobów restauracji. Zapewniłoby to, że rekomendacje są wykonalne i optymalizują wydajność operacyjną. Projekt skutecznie łączy różne komponenty techniczne w celu stworzenia kompleksowego systemu zarządzania, odzwierciedlając użyteczność i innowacyjność oczekiwaną w rozwiązaniach inżynieryjnych.

**Technical Language Precision:**

4: High Quality – The language is appropriate for a technical report. Terminology is used correctly, and statements are precise, with only minor shortcomings that do not affect the overall clarity.

---

### Official Review · Reviewer_BdxU · 2024-12-06
**Eatify System zarządzania restauracją**

**Confidence:** 5
**Significance Of Results:** 1
**Overall Quality:** 2

**Compliance With Template:**

4: High Quality – The article contains all the required sections, which are well-written and substantively correct, although minor errors or shortcomings may be present. The overall structure is clear and coherent.

**Description Of Results:**

1: Very Low Quality – The results are either not described or described in a minimal, unclear manner, without any examples or evidence. No evaluation is provided.

**Feedback On Consistency:**

Opis jest jednak dość chaotyczny. Nie zdefiniowano wszystkich użytkowników systemu, nie określono ich roli w systemie, nie wskazano co będzie ta aplikacja usprawniać w funkcjonowaniu restauracji, nie wskazano jakie ciekawe procesy będą zaimplementowane w tej aplikacji itd.

**Potential For Development:**

W przedstawionym opisie wzmiankowane są różne pomysły, natomiast są one bardzo dalekie od uzyskanych wyników.
Zrealizowane przedsięwzięcie jest na bardzo słabym poziomie i w tej postaci nie nadaje się praktycznie do dalszego rozwijania. Przede wszystkim należy rozpoznać specyfikę zarządzania restauracją i w tym celu powinno się podjąć chociaż najmniejszą próbę kontaktu oraz bezpośredniej i bieżącej współpracy z  menadżerem/właścicielem restauracji.

**Project Nature Evaluation:**

Zarządzanie restauracją to zagadnienie bardzo obszerne, a wykonanie systemu zarządzania restauracją nie jest odpowiednim zadaniem dla  czterech studentów w okresie 10 tygodni zajęć dydaktycznych. Ponadto studenci - niestety - zupełnie nie rozumieją, gdzie są problemy w zarządzaniu restauracją. Na stronie 2 napisali, że "Najważniejszym użytkownikiem systemu jest klient". Klient jest elementem takiego systemu, ale nie kluczowym dla zarządzania restauracją. Ich przedsięwzięcie zostało właściwie ograniczone do zamawianiu posiłków, Jeśli do tego ma się sprowadzić system zarządzania restauracją, to każdy, a najlepiej nawet bardzo mały, system sprzedaży, stosowany w obiektach handlowych, spełni te zadania, dodatkowo zapewniając elektroniczne płatności i także fakturowanie.
Skoncentrowano się głównie na takich problemach jak logowanie, szyfrowanie danych, płatności on-line, czy numerowanie stolików QR kodami (tylko po co?) pokazuje wyraźnie brak wiedzy na temat potrzeb osób zarządzających restauracjami. Ta wiedza nie może być pozyskana jedynie na podstawie analizy innych aplikacji do rezerwacji stolików i zamawiania dań w restauracji, bo jest to jedynie właśnie perspektywa klienta restauracji zbliżona do aktywności kupującego w sklepie internetowym. A jest to bardzo wąski wycinek zarządzania restauracją.

**Technical Language Precision:**

3: Average Quality – The language is mostly appropriate but may contain minor terminological or stylistic errors. Some statements might lack precision or require improvement for better readability.

---

### Official Review · Reviewer_YXgL · 2024-12-06
**Eatify - recenzja**

**Confidence:** 5
**Significance Of Results:** 4
**Overall Quality:** 5

**Compliance With Template:**

5: Very High Quality – The article contains all the required sections, which are written in a very detailed, clear, and error-free manner. The structure is professional and meets expectations, and the content adheres to the highest substantive and formal standards.

**Description Of Results:**

5: Very High Quality – The results are described in detail, clearly and comprehensively, supported by thorough evaluation, analysis, and convincing usage examples. The description meets the highest substantive standards.

**Feedback On Consistency:**

Abstrakt odznacza się spójnością oraz logicznym ułożeniem oraz powiązaniem składowych.

Wątpliwość, która miała jednak wpływ na zaniżenie oceny w sekcji "Significance Of Results" bierze się z dwóch konfliktowych stwierdzeń.
W sekcji "Prace związane z tematem" autorzy stwierdzają, że istniejące systemy cateringowe jak np. Glovo, Pyszne.pl, czy Wolt, obciążają restauratorów wysokimi prowizjami,a następnie, w tej samej sekcji piszą, że "Aplikacja Eatify jest odpowiedzią na potrzeby restauracji, które nie mogły być zaspokojone przez wcześniej wspomniane rozwiązania". Dalej, w sekcji "Wyniki" czytamy, że "Obsługuje ona zarówno zamówienia na wynos, jak i na miejscu". Jednakże, sekcja "Kierunki rozwoju" jasno podkreśla, że "(...) planowane jest także dostosowanie się do obecnych realiów i obsłużenie możliwości dostawy posiłków (...)". Stawia to w wątpliwość realizację wszystkich kluczowych funkcjonalności systemu i dlatego wpłynęło na ocenę.

Jeśli jest to nieprecyzyjne sformułowanie, to należy skupić się na jaśniejszym postawieniu i podkreśleniu celów projektu oraz rozróżnieniu ich od cech innych rozwiązań, czy problemów dziedzinowych. Pomóc może też lepsze, pod kątem formatowania tekstu, pogrupowanie fragmentów tematycznych - wstawienie przerw w "ścianę tekstu".

**Potential For Development:**

Wspomniane w abstrakcie możliwości rozwoju wydają się być adekwatne i dobrze dobrane - większość współczesnych firm stawia na tworzenie dedykowanych chatbotów, a wspomniana możliwość dostosowania szaty graficznej strony do stylu restauracji prosi się o bycie wręcz funkcjonalnością priorytetową. Nie mniej jednak, mając na uwadze ograniczenia czasowe oraz nacisk stawiany na bardziej kluczowe problemy samej branży, jasnym jest, że takiego rodzaju ograniczenia funkcjonalności musiały zostać przyjęte.

**Project Nature Evaluation:**

Projekt ewidentnie spełnia wymagania odnośnie prac inżynierskich dzięki realizacji typowego zadania za pośrednictwem dostępnych rozwiązań.

Należało by się jednak zastanowić, czy forma aplikacji webowej jest adekwatna do każdej sytuacji, np. gdy zamówienie miało by odbywać się już na sali. Wydaje się, że wersja mobilna mogła by być do takiej sytuacji lepiej dostosowana.

Dodatkowo, stwierdzenie (sekcja "Wniosków"), że "produkt jest w stanie obsłużyć zarówno małe lokale gastronomiczne, jak i duże restauracje" wydaje być się częściowo prawdziwa ze względu na metodę zastosowaną w systemie rekomendacyjnym tj. algorytm apriori, który skaluje się wykładniczo z powodu analizowania całego zbioru potęgowego przeglądanych opcji. Może generować duży narzut czasowy i pamięciowy dla większych baz danych. Skalowalność w sensie zwielokratniania wydaje się być zapewniona poprzez wykorzystane technologie.

Mając jednak na uwadze, że projekty inżynierskie z natury mogą wymagać dopracowania, nie wydaje się, aby pierwsza wersja propozycji miała ucierpieć z powodu nieefektywności.

W pracy rzuca się też w oczy mocne postawienie na prywatność poprzez zastosowanie szeregu technik kryptograficznych, które, w kontekście rozwiązywanego problemu, mogą być przesadzone. Nie oznacza to jednak, że same w sobie są czymś negatywnym dla projektu, a bardziej jego mniej istotną cechą.

**Technical Language Precision:**

5: Very High Quality – The language is entirely appropriate for a technical report. All terms are used correctly and precisely, and the style is professional, clear, and coherent, without any errors or ambiguities.

---

### Decision · Program_Chairs · 2024-12-10

Accept (Poster)